# Total Dietary Antioxidant Intake Including Polyphenol Content: Is It Capable to Fight against Increased Oxidants within the Body of Ultra-Endurance Athletes?

**DOI:** 10.3390/nu12061877

**Published:** 2020-06-23

**Authors:** Aslı Devrim-Lanpir, Pelin Bilgic, Tuğba Kocahan, Gökhan Deliceoğlu, Thomas Rosemann, Beat Knechtle

**Affiliations:** 1Department of Nutrition and Dietetics, Faculty of Health Sciences, Istanbul Medeniyet University, 34862 Istanbul, Turkey; asli.devrim@medeniyet.edu.tr; 2Department of Nutrition and Dietetics, Faculty of Health Sciences, Hacettepe University, 06100 Ankara, Turkey; pbilgic@hacettepe.edu.tr; 3Sport Medicine Physician, Department of Health Services, Sports General Directorship, The Ministry of Youth and Sports, Center of Athlete Training and Health Research, 06100 Ankara, Turkey; tugba.kocahan@sgm.gov.tr; 4Sport Scientist, Faculty of Sports Science, Kırıkkale University, 71450 Kırıkkale, Turkey; deliceoglugokhan@kku.edu.tr; 5Institute of Primary Care, University of Zurich, 8091 Zurich, Switzerland; Thomas.rosemann@usz.ch

**Keywords:** ultra-endurance, dietary antioxidants, total antioxidant capacity, exercise intensity, post-exercise recovery, 8-iso prostaglandin F2a

## Abstract

The role of dietary antioxidants on exhaustive exercise-induced oxidative stress has been well investigated. However, the contribution of total dietary antioxidant capacity on exogenous antioxidant defense and exercise performance has commonly been disregarded. The aims of the present investigation were to examine (i) the effects of dietary total antioxidant intake on body antioxidant mechanisms, and (ii) an exhaustive exercise-induced oxidative damage in ultra-endurance athletes. The study included 24 ultra-marathon runners and long-distance triathletes (12 male and 12 female) who underwent an acute exhaustive exercise test (a cycle ergometer (45 min at 65% VO_2_max) immediately followed by a treadmill test (75% VO_2_max to exhaustion). Oxidative stress-related biomarkers (8-isoprostaglandin F2alpha (8-iso PGF2a), total oxidant status (TOS, total antioxidant status (TAS)) in plasma were collected before and after exercise. Oxidative stress index was calculated to assess the aspect of redox balance. Blood lactate concentrations and heart rate were measured at the 3rd and 6th min after exercise. Dietary antioxidant intake was calculated using the ferric reducing ability of plasma (FRAP) assay. Dietary total antioxidant intake of the subjects was negatively correlated with pre-exercise TOS concentrations (rs = −0.641 in male, and rs = −0.741 in females) and post- vs. pre- (∆) 8-iso PGF2a levels (rs = −0.702 in male; *p* = 0.016, and rs = −0.682 in females; *p* = 0.024), and positively correlated with ∆ TAS concentrations (rs = 0.893 in males; *p* = 0.001, and rs = 0.769 in females; *p* = 0.002) and post- exercise lactate concentrations (rs = 0.795 for males; *p* = 0.006, and rs = 0.642 for females; *p* = 0.024). A positive meaningful (*p* = 0.013) interaction was observed between time at exhaustion and dietary antioxidant intake (rs = 0.692) in males, but not in females. In conclusion, the determination of total dietary antioxidant intake in ultra-endurance athletes may be crucial for gaining a better perspective on body antioxidant defense against exhaustive exercise-induced oxidative stress. However, the effects of dietary antioxidant on exercise performance and recovery rate needs further investigation.

## 1. Introduction

Ultra-endurance sports such as ultra-marathon races, ultra-cycling, and ultra-triathlon events requires athletes to push their limits, and to perform beyond their capacity for a long period of time [1,2]. It is known that moderate physical activity has quite beneficial effects for the body redox status which is playing a crucial role in exercise adaptation and cell-signaling [3], including a regulatory role in muscle regeneration during muscle damage [4], exercise-induced adaptations of muscle phenotype [5], and activation of the transcriptional factors like sirtuin-1 (SIRT-1) [6]. However, vigorous exercise results in a manifest increase that overwhelms the body antioxidant defenses [5]. It is well-established that exercise-induced oxidative stress caused by prolonged or exhaustive exercise has detrimental effects on skeletal muscles [7], fatigue [8,9], and immune function [10], which could all alter exercise performance.

A lot of research in recent years has focused on exogenous antioxidant supplements’ effect on achieving the balance between reactive oxygen species (ROS) and antioxidants in endurance athletes [11,12,13,14]. The acute supplementation of the well-known dietary antioxidants such as beta-carotene, vitamin C, vitamin E, selenium, coenzyme Q, thiols, and polyphenols are attracting widespread interest in fighting exercise-induced oxidative stress and enhancing performance [13,15,16,17,18]. However, some studies investigating the effects of the supplementations on the body redox status demonstrated that they showed no benefit due to their detrimental effects on decreasing both cell and muscle adaptation against exercise-induced oxidative stress and interfering with the beneficial influences of exercise [19,20]. Furthermore, some studies pointed out that the chronic supplementation may blunt the free radical-induced stress adaptation pathways in exhaustive exercise by interfering with the antioxidant defense against ROS within body systems [19,21,22]. Few studies have addressed that the antioxidant supplementation may be advantageous in athletes where the exogenous antioxidant levels are already depleted [12,23]. Therefore, before a supplementation to athletes should be considered, it is crucial to first determine whether the body antioxidant level is normal, and whether dietary antioxidant intake is needed to assess as well.

Oxidative stress-related research indicated that endogenous antioxidant defense is quite complicated, and is arranged by several antioxidant defense mechanisms affecting the body redox status [12,24,25,26,27]. Dietary antioxidants may be crucial for gaining a better perspective on body antioxidant levels. It is well-documented that antioxidant-rich foods provide several bioactive food constituents called phytochemicals that help to eliminate ROS and DNA damage by acting as an inducer of antioxidant defense mechanisms in the body [28,29]. Despite the fact that antioxidant rich-foods such as fresh and dry fruits, vegetables, nuts, and seeds are considered to play a major part in defending exercise-induced oxidative stress [29], studies commonly just focused on dietary micronutrient content disregarding polyphenolic functions, and few researchers have addressed the effects of antioxidant-rich foods on body redox status in athletes [23,30]. Thus, the dietary antioxidant intake effects on endogenous defense mechanisms and exercise performance in ultra-endurance athletes largely remain equivocal.

Numerous oxidative damage biomarkers have been identified during exhaustive or long-period exercise [31], however, the detection of them is mostly limited due to several factors including the lack of sensitivity, the low quantity, the very short half-life, the rapid interaction with antioxidants, the inapplicability of direct detection methods because of the instability, or the need for expensive devices and equipment [32,33]. Therefore, the difficulties in measuring different oxidant molecules, plasma total oxidant status (TOS), and total antioxidant status (TAS) is preferred to define the redox status under certain conditions such as exhaustive performance in athletes [23,34]. In addition to that, since exercise-induced ROS production is well-known to provoke lipid peroxidation, 8-iso prostaglandin F2alpha (8-iso PGF2a) is detected as a stable end product of the arachidonic acid, and defined as a suitable indicator of in vivo lipid peroxidation [35]. Several studies reported a parallel rise in the production of 8-iso PGF2a in plasma or urine, due to the accumulation of lipid peroxidation along with the increase in the intensity of the exercise [34,36,37,38].

In this study, we aimed to investigate the hormetic role of the dietary antioxidants, and to examine the interaction between the dietary antioxidant intake and the acute exhaustive exercise-induced oxidative stress in ultra-endurance athletes. It was hypothesized (i) that diets rich in antioxidants may increase the body exogenous antioxidant capacity after an exhaustive exercise. Furthermore, we assumed that a high dietary antioxidant consumption may also improve (ii) exercise performance by increasing the time until exhaustion during exercise and (iii) recovery, assessed by post-exercise heart rate and lactate concentrations in ultra-endurance athletes.

## 2. Materials and Methods

### 2.1. Study Participants

In total, 12 male (triathletes, *n* = 6; ultra-marathoners, *n* = 6; age 38.5 (31.3–40.0) years; body mass 72.6 (69.6–81.0) kg; body height 179.0 (173.5–184.5) cm; fat mass percentage 12.9 (10.0–16.3)%, fat-free mass 63.8 (60.0–68.3) kg) and 12 female (triathletes, *n* = 6; ultra-marathoners, *n* = 6; age 38.0 ((31.6–44.5) years; body height 162.0 (160.0–166.5) cm; fat mass percentage 19.4 (9.2–12.9)%; fat-free mass 45.5 (42.8–47.9) kg) were recruited from Ancyra Sports Club and local ultra-endurance groups by using study brochures and social media. Inclusion criteria for the study were: age 20–64 years old, being in a good health, training at least 15 h per week, not having any metabolic disease, being a non-smoker, participating at least one ultra-endurance race/event, taking no vitamins, minerals, dietary supplements, antibiotics, and any medication at least during the three months before the study. The lower age of participation was defined by age groups of triathlon and ultra-marathon races. Females with a regular menstrual cycle of physiological length (24–35 days) were included in the study. In addition, females in menopause or using oral contraceptives were excluded from the study. In literature, the influence of menstrual cycle on endurance performance is still controversially discussed. Whilst some studies found no interaction between different menstrual phases and maximal anaerobic performance [39], maximal oxygen consumption, cardiorespiratory variables and blood lactate concentration [40], few studies suggested that some interactions, i.e., VO_2_max was 2% lower in the luteal phase compared to the early follicular phase, endurance performance and muscle glycogen content were enhanced in the luteal phase. There is no consensus suggesting a specific menstrual phase for use in research on athletic performance. Therefore, to standardize the menstrual phase within subjects [41] and the estrogen effects on body antioxidant capacity by binding estrogen receptors and expand the antioxidant enzyme expressions using intracellular signal pathways [42], and to eliminate the effects of different phases on exercise performance [43], all females performed all exercise tests during mid-follicular phase (low estrogen levels, 7–9 days of the menstrual cycle). Menstrual cycle was calculated starting the first day of their period.

Subjects were informed of all potential risks involved and signed a written informed consent form before participating in the study. The study was performed at the Center of Athlete Training and Health Research of the Ministry of Youth and Sports, and was approved by Hacettepe University Ethics Board and Commission (research ethic project no, KA-180011) and The Ministry of Health (research ethic project no, E182).

### 2.2. Study Design

Subjects were required to visit the exercise laboratory on two visits. Before the visits, the subjects were informed to attend the laboratory at 8.00 a.m. to 9.00 a.m. after an overnight fast, and to refrain from exhaustive or long-duration exercise on the day preceding the visitations.

At the first visit, anthropometric parameters (body height, body weight, fat mass percentage, fat-free mass) were recorded, maximal oxygen uptake (VO_2_max) was measured, and a study questionnaire including quantitative food frequency questionnaire was applied to the subjects.

At the second visit, the study exercise test (a cycle ergometer (45 min at 65% VO_2_max) and immediately followed by a treadmill test (75% VO_2_max to the exhaustion) was performed, and blood samples were collected immediately before and after the test for analysis of oxidative stress (plasma 8-iso PGF2a) and total oxidant and antioxidant biomarkers, and blood lactate (La^-^) and heart rate were measured before, 3 and 6 min post-exercise to determine exercise performance and post-exercise recovery. Before both the VO_2_max test and the study exercise test, subjects were required to consume a standardized breakfast (50 g of carbohydrate, 6 g of protein, 3.5 g of fat) 2 h before the tests.

### 2.3. Dietary Records Assessment and Dietary Total Antioxidant Capacity Analysis

Dietary intake was determined using a validated quantitative food frequency questionnaire (QFFQ) [44] and 3-day food records. The QFFQ comprised 110 food items, and was used to assess the amount and frequency of total dietary consumption. The QFFQ was applied through face-to-face interviews by a sports dietitian. A photographic Meal and Food Atlas was used to record the quantity and portion size of food items more accurately [45]. The total dietary antioxidant intake was calculated using the intake frequency and total amount of each food item in QFFQ. The Ferric-reducing ability of plasma (FRAP) was calculated for each subject to assess the total antioxidant content of the diet [29]. To determine the dietary FRAP score, the Antioxidant Food Database developed by Carlsen et al. [29], which involves more than 3100 foods, spices, beverages, and herbs, was used. The FRAP assay measures the ability of reduction potential of each food to reduce ferric (Fe+3) ions to ferrous (Fe+2) ions. The FRAP assay was preferred because the assay does not include glutathione content of foods compared to other assays. This is considered an advantage because glutathione abundant in foods, but it is highly degraded in the human intestine and poorly absorbed in the body. FRAP scores were calculated with the exclusion of coffee in this study, in consistent with literature. Although the main antioxidant content of coffee has reported to be the Maillard products of the coffee which occurs during the roasting process, due to their high molecular weight, the percentage of absorption remains questioned. Therefore, coffee was excluded from total FRAP score to eliminate its effect on obscuring any other interactions between other sources of antioxidants and exercise-induced oxidative stress due to its high antioxidant capacity, in accordance with the literature [46].

An in-depth 24-h dietary recall was collected at the first visit to illustrate how subjects would collect the 3-day food record themselves. Before the second visit, the subjects completed 3-day food records (two working days and one weekend day). The food records data were converted to total daily energy and nutrient intake using the Nutrition Information System Software (BEBIS 6.1) program. When cooked meals were taken into account, the number of vitamins such as ascorbate, folate was adjusted by the software, i.e., the ascorbate was decreased to 50% of the uncooked value. Dietary micronutrient and micronutrient intakes were determined using Recommended Dietary Allowance (RDA) values approved by the National Institutes of Health (NIH) [47].

### 2.4. Maximal Oxygen uptake (VO_2_max) Measurement

Since the study included both ultra-marathon runners and triathletes, and since it has been reported that triathletes who have particularly trained for a triathlon show a similar VO_2_max for running and cycling [48], we preferred to measure VO_2_max on a treadmill. Before the VO_2_max test started, a warm-up of 5 min was performed at 5 km·h^−1^ and 0% gradient on the treadmill. After 5 min of passive recovery, the incremental running protocol started at 10 km·h^−1^ and with 0% gradient with the speed increasing by 1 km·h^−1^ each min until a running speed of 18 km·h^−1^ was achieved. After reaching and spending 1 min at the speed of 18 km·h^−1^, whilst the speed was fixed at 18 km·h^−1^, the gradient was increased by 1% each minute until exhaustion [49]. Heart rate was measured continuously and recorded at intervals of 30 s (Garmin HRM Soft Premium, Olathe, KS, USA). Respiratory parameters were recorded every 30 s (COSMED K5 metabolic cart; COSMED, Rome, Italy). Rating of perceived exertion (RPE) was used to monitor exercise intensity and fatigue during exhaustive exercise. The BORG 6-20 Category Scale was used to measure RPE by asking the subjects how tough they felt during an exercise bout at the end of each stage until volitional exhaustion [50].

VO_2_max was determined as the highest average of the two highest sequential reading, in the case at least two of the following criteria was observed: (1) Whilst the exercise load increased, VO_2_ remained stable in the last two stages (range < 2.1 mL·kg^−1^·min^−1^), (2) Respiratory exchange ratio (RER) ≥1.10, (3) maximum heart rate (HRmax) >90% of age-predicted HRmax (220-age) and (4) blood lactate concentration ≥8.00 mmol·L^−1^ [51].

### 2.5. Exercise Protocol

Subjects were initially cycling for 45 min at a submaximal speed (65 ± 5% VO_2_max), and then immediately followed a running test at 75% VO_2_max to exhaustion defined as inability to maintain running intensity. The expiration parameters (COSMED K5 metabolic cart; COSMED, Rome, Italy), and heart rate (Garmin HRM Soft Premium, Olathe, KS, USA) were recorded all the exercise period. The exercise test was performed under controlled laboratory conditions (20–25 °C, 40% relative humidity). All athletes were familiarized with cycling on a cycle ergometer and running on a treadmill.

The study protocol was determined based on Montenegro et al. [52], and was slightly changed based on our study aim. The time-to-exhaustion (TTE) exercise protocol was chosen because it has been applied to a similar population, and it has shown a significant decrease in response to exercise intensity [23]. Watson et al. [23] reported that the intensity of TTE exercise under 80% VO_2_max appeared to have a lower coefficient of variation. According to that, the exercise intensity of the study was arranged at 75% VO_2_max to exhaustion. The protocol ended with verbal approval by athletes, stating that they could not maintain the exercise intensity. The total running time was recorded.

### 2.6. Exercise Performance Measurement

Exercise time-to-exhaustion and post-exercise recovery of blood lactate and heart rate parameters were determined as exercise performance measurements.

### 2.7. Plasma Total Oxidant Status (TOS) and Plasma Total Antioxidant Status (TAS) Analysis

Two 18 mL venous blood parameters were collected into EDTA coated tubes immediately before and after the study exercise protocol. Plasma pellets were obtained by centrifugation at 3000× *g* for 10 min at 4 °C. Plasma total oxidant status was analyzed using an automated method (Total Oxidant Status Kit, Rel Assay^®^ Diagnostics, Ankara, Turkey). The method is based on the oxidation of Fe^+3^ to Fe^+2^ in the presence of reactive oxidants in the acidic medium. The TOS results were expressed in μmoL H_2_O_2_ equivalent/L (μmoL H_2_O_2_ eq/L). Plasma total antioxidant status was measured using the colorimetric test system (Total Antioxidant Status Kit, Rel Assay^®^ Diagnostics, Ankara, Turkey). The calibration of reaction was performed using Trolox (a water-soluble analogue of vitamin E, 6-hydroxy-2.5.7.8-tetramethylchroman-2-carboxylic acid), and the total antioxidant status was expressed as μmoL Trolox equivalent/L (μmoL Trolox eq/L).

#### Oxidative Stress Index (OSI) Calculation

Plasma TOS to TAS ratio was defined as the oxidative stress ındex (OSI) [53]. Before calculation, the TAS unit expressed in μmoL has to be converted to mmol, and calculated by the following formula;

OSI (arbitrary unit) = TOS (μmoL H_2_O_2_ equivalent/l)/TAS (mmol Trolox equivalent/l).

### 2.8. Plasma 8-isoprostaglandin F2alpha (8-iso PGF2a) Analysis

To assess the exercise-induced oxidative stress, 8-iso PGF2a level in plasma was measured using a competitive enzyme-linked immunosorbent assay kit (Elabscience^®^ Biotechnology Co., Ltd., Houston TX, USA) (sensitivity: 9.38 pg/mL, detection range from 15.63 to 1000 pg/mL).

### 2.9. Lactate Analysis

To our knowledge, there is no certain suggestion to determine lactate clearance after exhaustive exercise, and its measurement 2 or 3 min after exercise is generally preferred in literature. In addition, Gass et al. [54] conducted a study to determine blood lactate concentration following a maximal exercise in trained athletes and stated that peak lactate values after maximal exertion was reached 6 min after exercise. Therefore, we preferred to measure the blood lactate concentration at the end of each 3 min and after exercise at 3 and 6 min to measure how diet in rich antioxidants effects on post-exercise lactate removal, especially at the peak lactate concentration time as it was practiced by Oh et al. [55] similarly in determining the removal of lactate after high-intensity exercise, and by Di Masi et al. [56] in comparing blood lactate clearance performed during cycling in water immersion and during cycling on land after a similar exercise bout. Blood lactate levels were measured using a validated portable blood lactate analyzer (Lactate plus, Nova Biomedical, Waltham, MA, USA). The quality control solutions provided by the manufacturer were used prior to testing, and a blood sample of 0.7 μL was required to assess the lactate concentration in capillary blood.

### 2.10. Statistical Analysis

Statistical analysis was performed using IBM SPSS Statistics Software for Windows version 23.0 (IBM Corporation, Armonk, New York, NY, USA). Sample size was calculated by assuming that 8-iso PGF2a levels would increase 70% after exhaustive exercise (Mrakic-Sposta et al.) [37,57], a total of 24 subjects (12 males, 12 females) were required. Prior to analysis, data were tested for normality using the Kolmogorov- Smirnov test, and were found to be non-normally distributed. Therefore, all data were reported as median and interquartile range. The Wilcoxon rank test was used to compare pre- and post-exercise plasma parameters. Effect size (*d*) for non-parametric tests was calculated using the following formula: r = z/√N [58]. r is referred to as *d*, z is referred to as z value of Wilcoxon rank tests, N is referred to as the total number of subjects. To determine effect sizes; *d* < 0.2 was classified as a small effect, *d* between 0.2 and 0.5 was considered as a medium effect, and *d* > 0.8 was considered as a large effect. The Spearman’s rho correlation coefficient was performed to measure the strength of an interaction between dietary antioxidant capacity (calculated FRAP score), and pre- and post-exercise changes of plasma oxidative stress, total oxidant and antioxidant parameters, post-exercise changes of heart rate (3rd HR–6th HR), lactate (3rd La^-^–6th La^-^), and TTE exercise, adjusted for years of training and average training to prevent their mixing effects on the outcome variables. All data were set at the 5% level (*p* < 0.05).

## 3. Results

### 3.1. Subjects

Basic characteristics and dietary intake based on sex of subjects are displayed in Table 1. No difference was observed according to their body weight (per kg) from carbohydrate (*p* = 0.085) and protein (*p* = 0.124) between sexes. The intakes met or exceeded the RDA values with the exception of vitamin E (94.4% of the requirement was met in females), vitamin A (81.0% of the requirement was met in males, and 72.1% of the requirement was met in females), and selenium (95.0% of the requirement was met in females).

### 3.2. Dietary Antioxidant Intake

The amount of dietary antioxidant intake for total and each food group is presented in Figure 1. Including with the contribution of coffee, the coffee comprised 38.5% and 44.6% of total FRAP scores in males and females, respectively. Due to coffee’s possible effects on any obscure interactions between other antioxidant sources of FRAP and exercise performance, the total antioxidant intake was assessed excluding the coffee. Foods were grouped as vegetables, fresh fruits, dry fruits, cereals, legumes, nuts and seeds, dairy, meats, eggs, and beverages. No statistical difference was found with regard to the antioxidant capacity for each food group between sexes (sum of FRAP; 16.6 (14.8–22.8) mmol/day and 17.2 (14.1–23.6) mmol/day, males and females respectively) (*p* = 0.908).

### 3.3. Changes in Plasma Oxidative Stress Biomarkers after Exercise Testing

Table 2 summarizes the role of acute exhaustive exercise on changes of plasma oxidative stress biomarkers. All oxidant and antioxidant biomarkers indicated a significant effect of the exhaustive exercise. Plasma 8-iso PGF2a levels increased up to 444.6 (410.6–482.3) pg/mL from 230.7 (217.9–305.0) pg/mL in males (*d =* 0.60, *p* = 0.003), and up to 458.01 (447.2–556.0) pg/mL from 223.8 (214.2–268.3) pg/mL in females (*d =* 0.60, *p* = 0.003) and the calculated oxidative stress index increased up to 0.4 (0.3–0.4) from 0.3 (0.3–0.4) in males (*d =* 0.63, *p* = 0.002), and up to 0.3 (0.3–0.4) from 0.3 (0.2–0.3) in females (*d =* 0.63, *p* = 0.003) after exercise. The significant increase was also found for total antioxidant parameters (median changes in plasma TAS levels; 13.6 % for males (*d =* 0.62; *p* = 0.002) and 12.0 % for females (*d =* 0.63; *p* = 0.002)). The calculated oxidative stress index suggested an increase after exercise (median changes in OSI; 27.5 % for males (*d =* 0.5, *p* = 0.002) and 20.7 % for females (*d =* 0.6, *p* = 0.003)).

### 3.4. Dietary Antioxidants and Exercise Performance

Table 3 represents the interaction between total dietary antioxidant capacity and both plasma exercise-induced redox status and oxidative stress biomarkers. The years of training and average training time were adjusted to eliminate their confounding effects on the results. Dietary antioxidant intake was negatively correlated to pre-exercise TOS concentrations (rs = −0.641 for males; *p* = 0.025, and rs = −0.741 for females; *p* = 0.006) and ∆(post- vs. pre-) 8-isoPGF2a concentrations (rs = −0.702 for males; *p* = 0.016, and rs = −0.682 for females; *p* = 0.024). A positive correlation between ∆TAS concentrations was observed (rs = 0.893 for males, and rs = 0.769 for females; *p* = 0.001 and *p* = 0.002 respectively).

The difference between ∆TOS concentrations was not significantly related to FRAP scores (*p* = 0.216 for males, and *p* = 0.101 for females). The median running time was 26.7 (26.0–28.1) min in males and 26.5 (25.6–27.2) min in female subjects. A positive meaningful (*p* = 0.013) interaction was observed between TTE exercise (rs = 0.692) for males. The median lactate concentration 3 and 6 min after exercise was 12.2 (10.9–13.5) mmol/L and 10.40 (9.5–11.9) mmol/L in males and 10.2 (9.3–11.2) mmol/L and 10.1 (8.6–10.4) mmol/L in females, respectively, and pointed out that the high intensity of the exercise protocol applied in the study. The post-exercise ∆lactate concentrations (rs = 0.795 for male athletes, and rs = 0.642 for females; *p* = 0.006 and *p* = 0.024, respectively) were positively correlated to dietary FRAP score. The median heart rate 3 and 6 min after exercise was 146.0 (135.5–153.8) and 132.5 (127.3–147.8) bpm in males and 141.0 (133.3–154.8) and 132.0 (123.3–147.5) bpm in females. No significant relationship was observed between post-exercise ∆heart rate values and dietary antioxidant intake (*p* = 0.440 for males and *p* = 0.819). The median of RPE was 19.0 (18.3–20.0) in males and 19.00 (18.3–19.0) in females and indicated that the applied exercise test was perceived as quite exhausting by the subjects. RPE scores were not significantly related to dietary antioxidant intake (*p* = 0.079 for males and *p* = 0.945 for females).

## 4. Discussion

The main purpose of the present study was to investigate the effects of dietary antioxidant capacity on exhaustive exercise-induced oxidative stress and total oxidant/antioxidant status in ultra-endurance athletes. The main findings were that total dietary antioxidant capacity had a significant influence on body antioxidant defense against acute exhaustive exercise-induced oxidative stress. A diet high in antioxidants was positively related to blood lactate removal after exercise for both sexes, and running time to exhaustion just in males. However, no significant interaction was observed between dietary antioxidants and post-exercise HR recovery. Therefore, the hypothesis that consuming an antioxidant-rich diet would increase the body exogenous antioxidant defenses developed against extreme endurance exercise has been confirmed, but its effects on exercise performance and recovery remained uncertain.

In the present study, the dietary micronutrient analysis indicated that subjects mainly met or exceeded the daily recommended intake except for vitamin E for females and vitamin A for both sexes. Although the consumption of vitamin E and A seemed not problematic and the subjects mostly met the RDA [47], it should be taken into account that the RDA values are designed for non-athletic populations, and athletic populations may require more than a sedentary population, as demonstrated in literature [59]. Considering total dietary antioxidant intake calculated with FRAP, the median dietary antioxidant content was observed 16.6 (14.8–22.8) mmol/day in males and 17.2 (14.1–23.6) mmol/day for females. Carlsen et al. [29] provided no cut-off point for the FRAP score to determine total dietary antioxidant content high in antioxidants. However, Koivisto et al. [60] performed a randomized control placebo trial in elite endurance athletes to investigate the effects of dietary antioxidant intervention during a 3-week altitude training camp on improvement of oxygen-carrying capacity and exercise performance. The total antioxidant concentration of dietary intervention was calculated using the FRAP assay. The FRAP score of 22.2 mmol/day was defined as a diet high in antioxidant, and 2.8 mmol/day was defined as a low antioxidant diet. Comparing with their research [60], the subjects in the present study consumed a moderate-to-high dietary antioxidant diet. Thus, taking together with total dietary antioxidant and micronutrient intake, the subjects’ diet may meet the requirements to defense against exhaustive exercise-induced ROS.

In this study, we investigated the role of dietary antioxidants and polyphenolic content on oxidative stress, since research on oxidative stress often ignored its potential impact on oxidative metabolism and performance. Several dietary strategies were applied to diminish or protect against the exercise-induced oxidative damage that caused accumulation of excessive oxygen species, and had an adverse effect on muscle damage, contractile function, lipid peroxidation [34,61]. However, the studies were predominantly carried out by applying antioxidant supplementation, and the effects on oxidative stress remained controversial. Antioxidant micronutrients such as vitamins A, C, E, D, the B vitamins, zinc, beta carotene, and selenium have been commonly studied based on their antioxidant properties [11,13,14,15,28]. Mastaloudis et al. [14] performed a randomized control placebo trial in runners to investigate the effects of supplementation with 300 IU vitamin E and 1000 mg vitamin C for 6 weeks prior to a 50 km ultra-marathon. Plasma F2-prostanes were elevated just in the placebo group after the ultra-marathon, but inflammation biomarkers (Interleukin-6, C-reactive protein, tumor necrosis alpha) were increased regardless of treatment or placebo. The results pointed out that an antioxidant supplementation protects against exercise-induced oxidative stress, but not against inflammation after an ultra-marathon. McAnulty et al. [62] assessed the effects of vitamin E supplementation with 800 IU for two months prior to a triathlon race (a 2.4-mile ocean swim, a 112-mile bike race and a 26.2-mile run) on plasma homocysteine and oxidative stress biomarkers in triathletes. Cortisol was increased regardless of the treatment, and plasma F2-isoprostanes was significantly increased in the treatment group (181%) compared to the placebo group (97%) suggesting that a prolonged vitamin E supplementation exhibited pro-oxidant characteristics in triathletes during exhaustive exercise. Since the antioxidant supplementation may cause a detrimental effect on body redox metabolism, and research on the effects of dietary antioxidants on exercise performance revealed that antioxidant supplementation has only influenced exercise performance on athletes with clinically nutrient deficiencies [23], the need for understanding the crucial role of dietary antioxidant consumption comes into prominence.

The acute exhaustive exercise increased ROS production and plasma oxidative stress (change in 8-iso PGF2a levels; 80.5% for males, and 99.1% for females) and total oxidant biomarkers (change in TOS levels; 30.3% for males and 48.2% for females) were significantly increased after the exercise. Research on oxidative stress and performance suggested that an increase in 8-iso PGF2a served as a more reliable indicator to determine lipid peroxidation [23,34]. The increase of both plasma 8-iso PGF2a and TOS (as a biomarker of oxidative stress) and the oxidative stress index (as an indicator of increased oxidative status) found in this study are consistent with existing literature [34,37,38,53]. As commonly demonstrated, both exhaustive exercise and a long period of exercise are highly associated with a remarkable accumulation of oxidative radicals beyond the body’s limits to defense ROS [35]. Therefore, a detailed insight into the body’s antioxidant defense under exhaustive exercise becomes of the utmost importance.

The findings of a meaningful increase of plasma TAS (as a biomarker of antioxidant capacity) after exhaustive exercise test was in contrast to earlier studies [23,63]. A potential explanation for the observed results could be attributed to the body antioxidant defense capacity which was thought to improve with both training and adaptation to exercise [64]. The average years of training of the subjects were 6.0 (2.8–20.0) years for males, and 10.5 (4.0–20.0) years for females suggesting that the longer periods in performance could be an effective factor for the adaptation of the body towards oxidative stress.

It has been reported that a dietary intervention has a significant influence on the exercise-induced F2-isoprostanes concentrations [23]. In line with existing literature, a meaningful negative interaction was found between dietary antioxidant consumption and ∆8-iso PGF2a concentrations indicating that changes in oxidative stress after exercise arises depending of the dietary antioxidant capacity. Furthermore, a meaningful negative interaction between TOS at rest and dietary antioxidant intake (calculated using FRAP assay) was found. These results indicated that antioxidant-rich diet consumption may have a paramount effect on body antioxidant defense.

In the present study, no significant relationship was observed between plasma TAS at rest and diet FRAP scores. Plasma total antioxidant concentration at rest may not be a good predictor to determine the body antioxidant capacity because of its disregarding potential to consider individual antioxidant tissue stores and differences in mobilization ability of the stores in the plasma [34]. Therefore, the determination of the interaction between exercise-induced plasma TAS changes and FRAP scores could be a better choice to interpret the effects of dietary antioxidant intake on exercise-altered antioxidant mechanism of subjects. A positive relationship between exercise-induced changes of total plasma antioxidant capacity and FRAP scores also suggested that these ultra-endurance athletes consumed antioxidant-rich diets which could improve a better antioxidant defense against oxidative damage after exhaustive exercise.

In this study, a time-to-exhaustion exercise protocol was preferred because its effects on oxidative stress have already been confirmed in other studies, and treadmill-based TTE could be classified as a reliable research tool to determine both the endurance performance and the relative exercise intensity in trained athletes [23]. It is well-known that oxidative damage is increasing in line with both exercise duration and exercise intensity [8]. In addition to that, blood lactate concentration and heart rate scores monitored after exercise are also other indicators of exercise performance and post-exercise recovery, and commonly used in clinical practice [65,66,67]. It is well reported that lactate production gradually increases in line with exercise intensity during exhaustive exercise, and blood lactate clearance rate is a good predictor for the rate of post-exercise recovery [68,69]. Considering all the information related to both exercise performance and post-exercise recovery, a meaningful interaction was observed between dietary antioxidant capacity and TTE and ∆lactate concentrations in males. Dietary antioxidant capacity was positively related to ∆lactate concentrations in female subjects, but not related to lactate concentrations 3 and 6 min after exercise. All the results addressed that dietary antioxidant intake may affect exercise performance and post-exercise recovery in males. However, the same interaction cannot be confirmed for females when only considering lactate concentrations.

Our findings showed no significant interaction between the FRAP and RPE scores. RPE has been identified as a common indicator to describe how hard an exercise task is during exercise, and has been defined in terms of its strong interaction between exercise intensity [70,71]. Doherty and Smith [71] conducted a meta-analysis related to the role of caffeine on exercise capacity. In their study, the caffeine-mediated improvement of exercise capacity was assessed based on the reduced RPE levels after exercise. A similar interaction between RPE and performance was demonstrated in the study, in which the influence of nitrate and caffeine consumption on exercise performance was examined [72]. The possible explanation why no interaction was detected in our results was that the self-perceived fatigue during exercise was almost similar in all subjects (median 19 for both sexes) and therefore not related to dietary antioxidant consumption.

A limitation of the study was that ultra-endurance athletes compete under more extreme conditions compared to the study exercise protocol. However, we preferred a TTE protocol immediately after a 45 min cycle ergometer exercise with a submaximal speed to push the body’s limits, and thus to increase the exercise-induced oxidative stress in subjects. In addition, lactate accumulation concentrations, RPE and HR scores after exercise indicated the strenuousness of the exercise protocol. Future studies may be performed in a competitive ultra-marathon or a triathlon race to ensure more reliable conditions to assess the relationship between the dietary antioxidant intake and its effects on both the body redox balance and exercise performance. On the other hand, the main strength of our study was that this is the first study conducted to calculate total dietary antioxidant content by FRAP assay in ultra-endurance athletes. Furthermore, the determination of the all dietary records and analyses was applied by a qualified sports dietitian. This study provides the framework for future studies to consider the substantial role of dietary antioxidant capacity on both performance and exercise-induced oxidative damage. In future research on exercise-induced oxidative stress, the dietary records should also be investigated to provide more insight into its influence on oxidative balance in ultra-endurance athletes.

## 5. Conclusions

In conclusion, exhaustive exercise attenuated oxidative stress and altered redox balance in ultra-endurance athletes. We confirmed that total dietary antioxidant capacity had a significant role to assess the exogenous antioxidant defense of the body, therefore a diet rich in antioxidants may provide a better oxidative balance after an exhaustive exercise. Dietary antioxidant intake including antioxidant and polyphenolic content may positively improve both exercise performance and post-exercise recovery, however further work is needed. Our results are encouraging and should be validated in a larger cohort of ultra-endurance athletes.

## Figures and Tables

**Figure 1 nutrients-12-01877-f001:**
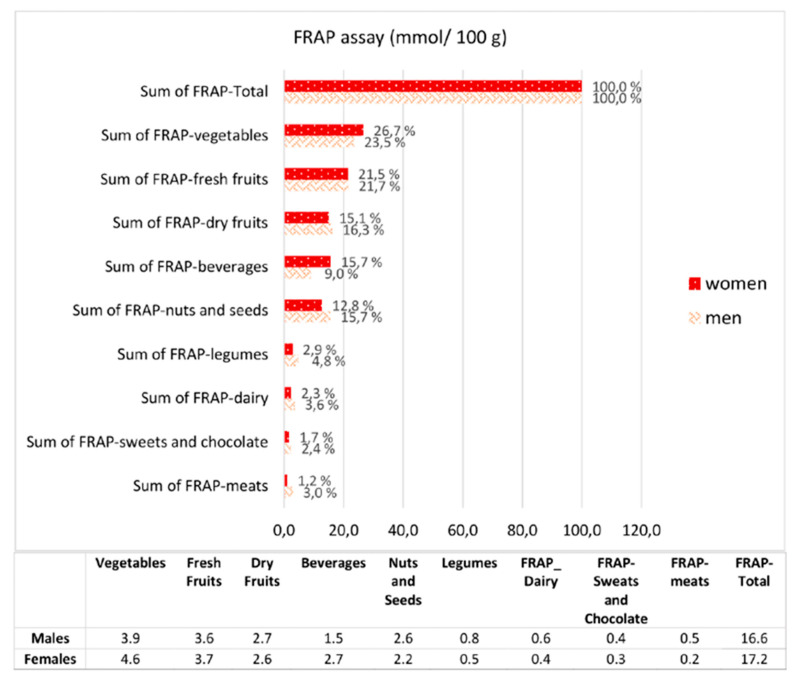
The contribution of foods and beverages to dietary total antioxidant intake according to FRAP assay [27].

**Table 1 nutrients-12-01877-t001:** Basic characteristics and dietary intake of subjects, median (Interquartile range).

Parameter	Males (*n* = 12)	Females (*n* = 12)
Age (y)	38.5 (31.3–40.0)	38.0 ((31.6–44.5)
Height (cm)	179.0 (173.5–184.5)	162.0 (160.0–166.5) *
Body mass (kg)	72.6 (69.6–81.0)	56.5 (53.5–59.0) *
Fat mass percentage (%)	12.9 (10.0–16.3)	19.4 (9.2–12.9) *
Fat-free mass (kg)	63.8 (60.0–68.3)	45.5 (42.8–47.9) *
Maximum oxygen consumption (VO_2_ max), mL·min^−1^·kg^−1^	60.7 (52.7–65.1)	51.0 (48.7–52.2) *
Baseline training (h·week^−1^)	16.3 (15.0–17.6)	16.4 (15.5–17.0)
Years in ultra-endurance sports (y)	6.0 (2.75–20.0)	10.5 (4.0–20.0)
**Dietary intake per day**	**%RDA ****	**%RDA ****
Energy (kcal)	2571.8 (2057.4–3355.5)	1871.7 (1589.7–2020.0) *
Carbohydrate (%)	36.1 (33.2–41.5)	34.0 (29.6–35.9)
Carbohydrate (g^−1^·kg^−1^·d)	3.4 (2.3–3.8)	2.7 (2.1–3.2)
Protein (%)	19.9 (16.7–21.8)	16.3 (14.8–18.2) *
Protein (g^−1^·kg^−1^·d)	1.6 (1.3–2.2)	1.3 (1.2–1.3)
Fat (%)	43.8 (40.9–49.9)	49.8 (48.9–52.9) *
Omega 3 (g)	2.9 (2.4–4.1)	2.6 (1.9–3.6)
Omega 6 (g)	25.4 (19.1–32.0)	20.0 (14.6– 24.5)
Vitamin C (mg)	145.6 (84.9–186.1) 161.7	98.9 (68.0–121.3) 136.7
Vitamin E (mg)	28.7 (20.8–31.7) 143.6	18.9 (12.5–25.2) * 94.4
Vitamin A (RAE)	729.2 (592.1–815.9) 81.0	505.0 (403.5–915.0) 72.1
Selenium (μg)	52.2 (43.4–65.0) 95.0	57.5 (45.2–62.1) 104.5
Zinc (mg)	14.8 (13.5–20.8) 134.5	10.6 (10.1–11.7) * 132.0

* *p* < 0.05; ** % RDA represents the percentage of median micronutrient intakes of subjects compared to the recommended dietary allowance (RDA).

**Table 2 nutrients-12-01877-t002:** Plasma oxidative stress parameters before and after the exercise protocol.

Plasma Parameters		Pre-Exercise	Post-Exercise	∆ (Post- vs. Pre-)	Effect Size (*d*)	Change (%; Range)	*p*
TOS ^a^ (μmol H_2_O_2_eq/L)	Men	3.9 (3.6–4.5)	5.0 (4.8–7.3)	1.1 (0.5–2.9)	0.50	30.3 (11.0–84.0)	0.015 *
	Women	3.4 (3.2–4.1)	5,0 (4.6–5.6)	1.7 (0.6–2.3)	0.59	48.2 (18.0–65.4)	0.004 *
TAS ^b^ (μmol Trolox equivalent/L)	Men	1.6 (1.5–1.6)	1.8 (1.7–1.8)	0.2 (0.2–0.3)	0.62	13.6 (11.1–16.1)	0.002 *
	Women	1.4 (1.3–1.5)	1.6 (1.5–1.7)	0.2 (0.1–0.2)	0.63	12.0 (9.5–15.9)	0.002 *
OSI ^c^	Men	0.3 (0.3–0.4)	0.4 (0.3 0.4)	0.1 (0.1–0.1)	0.63	27.5 (19.7–40.1)	0.002 *
	Women	0.3 (0.2–0.3)	0.3 (0.3– 0.4)	0.1 (0.0–0.1)	0.60	20.7 (14.1–36.2)	0.003 *
8-iso PGF2α (pg/mL)	Men	230.7 (217.9–305.0)	444.6 (410.6–482.3)	193.5 (171.0–228.0)	0.60	80.5 (58.7–112.6)	0.003 *
	Women	223.8 (214.2–268.3)	458.0 (447.2–556.0)	219.6 (179.0–290.7)	0.60	99.1 (58.6–143.4)	0.003 *

Concentrations are expressed as median (interquartile range). ^a^ TOS: total oxidant capacity (plasma); ^b^ TAS: total antioxidant capacity; ^c^ OSI: oxidative stress index.

**Table 3 nutrients-12-01877-t003:** Correlations of pre- and post- exercise plasma biomarkers related to oxidative stress with dietary antioxidant intake, adjusted for years of training, and average training.

		Dietary Antioxidant Intake (FRAP-mmol/day)			
		Males (*n* = 12)		Females (*n* = 12)	
		*r*	*p*	*r*	*p*
TOS^a^ (μmol H_2_O_2_eq/L)	Pre	−0.641	0.025 *	−0.741	0.006 **
	Post	0.147	0.648	−0.077	0.812
	∆(Post- vs. Pre-)	0.343	0.216	−0.497	0.101
TAS ^b^ (μmol Trolox equivalent/L)	Pre	0.225	0.483	0.077	0.812
	Post	0.514	0.087	0.417	0.178
	∆(Post- vs. Pre-)	0.893	0.001 **	0.769	0.002 **
OSI ^c^ (arbitrary unit)	Pre	0.320	0.311	−0.451	0.141
	Post	0.056	0.863	−0.077	0.811
	∆(Post- vs. Pre-)	−0.409	0.187	0.266	0.404
8-isoPGF2a (pg/mL)	Pre	−0.417	0.201	−0.373	0.259
	Post	−0.573	0.066	0.055	0.873
	∆(Post- vs. Pre-)	−0.702	0.016 *	−0.682	0.024 *
Time-to-exhaustion (min)		0.692	0.013 *	−0.028	0.931
Lactate (mmol/L)	∆(Post- 3rd La^-^–6th La^-^)	0.795	0.006 **	0.642	0.024 *
Heart rate (HR) (bpm)	∆(Post- 3rd La^-^–6th La^-^)	0.246	0.440	0.074	0.819
RPE ^d^	Post	−0.525	0.079	−0.022	0.945

Spearman’s rho correlation. * *p* < 0.05. ** *p* < 0.001; ^a^ TOS: total oxidant capacity (plasma); ^b^ TAS: total antioxidant capacity; ^c^ OSI: oxidative stress index ^d^ RPE: rating of perceived exertion.

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
