# Peer review of "Total Dietary Antioxidant Intake Including Polyphenol Content: Is It Capable to Fight against Increased Oxidants within the Body of Ultra-Endurance Athletes?"

_nutrients, 2020, doi:10.3390/nu12061877_

Round 1

Reviewer 1 Report

Interesting and well written study! Just a few questions to clarify findings:

  1. In the statistical methods section its reported all values are shown as mean and interquartiles. Table one says its median and interquartiles. Please clarify which and edit. 
  2. Calories seem quite low, especially for women exercising 16+ hours/week. Perhaps add to the discussion the limitations of using FFQs to extrapolate dietary assessments and the potential standard deviation of all nutritional values because of it. 
  3. When reporting Omega 3 and Omega 6, are those values in grams? please add unit of measure. 

Author Response

Reviewer 1

Interesting and well written study! Just a few questions to clarify findings:

In the statistical methods section its reported all values are shown as mean and interquartiles. Table one says its median and interquartiles. Please clarify which and edit. 

Answer: We agree with the expert reviewer. In the statistical methods, we wrote as “mean” instead of “median” by mistake. Because all values are shown as median in the paper, we corrected it as “median” in the statistical methods section (current line: 254).

Calories seem quite low, especially for women exercising 16+ hours/week. Perhaps add to the discussion the limitations of using FFQs to extrapolate dietary assessments and the potential standard deviation of all nutritional values because of it. 

Answer: We agree with the expert reviewer that calorie consumption found quite low in subjects, however we want to emphasize that all dietary assessments and macro- and micronutrient values are determined according to 3-day food records instead of FFQs (stated in line 151-158). However, we can definitely add the standard deviation of all nutritional values if you still suggest that adding standard deviation values is better to clarify dietary assessments.

When reporting Omega 3 and Omega 6, are those values in grams? please add unit of measure. 

Answer: We agree with the expert reviewer and added unit of Omega 3 and Omega 6 in table 1.

Reviewer 2 Report

The article presented is very interesting and provides a breakthrough in the knowledge of the use of antioxidants in sports performance. However, I have seen some limitations that limitations be corrected:

Introduction:

Line 51: Could you provide more information about the effects of physical activity on exercise adaptations or cell-signaling processes?

Line 61: Could the authors explain the reasons due to some studies reported no benefits on body redox status?

Material and methods:

Line 102: More information about anthropometric characteristics of the participants is required.

Line 105: Why the authors selected participants between 24-49 years? Please provide the rationale explanation. Did the authors found statistical differences between participants categories?

Line 108: Menstrual cycle phase was the same in all the women participants? How the authors controlled it during the experiments? Please explain carefully.

Line 116: Could the authors provide more information about the time-of-day when the experiments were realized?.

Line 119: The authors stated "At the first visit, anthropometric parameters (body height, body weight, fat mass percentage, fat-free mass) were recorded". Please provide more information about the measurements...

Line 162: Why the authors started the VO2max test at 5 km.h-1 and 0 % gradient on the treadmill?

Line 164: Typo "1 km.h-1" Please correct along the manuscript.

Line 170: The authors claim that  "BORG 6-20 Category Scale was used to measure perceived exertion after exercise". More data is needed about the time when the Borg Scale was applied.

Line 214: Why the authors selected 3-min and 6-min post-exercise for measuring blood lactate?.

Results:

Line 239: Please provide the exact p-value not p>0.05 in all the results section and provide two decimals in the effect sizes values.

Line 253: 16.6 (14.8-22.8) IC 95%? Please define.

Line 266: The authors should change Figure 1, the figure no it so clear...

Line 268: Some typos en Table 2.

Disccusion:

Lines 304-307: Please rewording.

Author Response

Reviewer 2

The article presented is very interesting and provides a breakthrough in the knowledge of the use of antioxidants in sports performance. However, I have seen some limitations that limitations be corrected:

Introduction:

Line 51: Could you provide more information about the effects of physical activity on exercise adaptations or cell-signaling processes?

Answer: We agree with the expert reviewer and provided more information about the effects of moderate activity on exercise adaptation and cell-signaling processes as follows: “…..Although it is known that moderate physical activity has quite beneficial effects for the body redox status which is playing a crucial role in exercise adaptation and cell-signaling [3] including a regulatory role on muscle regeneration during muscle damage [4], exercise-induced adaptations of muscle phenotype [5], and activation of the transcriptional factors like sirtuin-1 (SIRT-1) [6],…” (Current line: 51-53).

Line 61: Could the authors explain the reasons due to some studies reported no benefits on body redox status?

Answer: We agreed with the expert reviewer and explained the reasons why some studies highlighted no benefits of antioxidant supplementation on body redox status as follows: “….. However, some studies investigating the effects of the supplementations on the body redox status demonstrated that they showed no benefit due to their detrimental effects on decreasing both cell and muscle adaptation against exercise-induced oxidative stress and interfering with the beneficial influences of exercise ….” (Current Line: 63-66).

Material and methods:

Line 102: More information about anthropometric characteristics of the participants is required.

Answer: We agree with the expert reviewer and added more information about anthropometric characteristics of the subjects as follows: “…..In total, 12 male (triathletes, n=6; ultra-marathoners, n=6; age 38.5 (31.3-40.0) y; body mass 72.6 (69.6-81.0) kg; height 179.0 (173.5-184.5) cm; fat mass percentage 12.9 (10.0-16.3) %, fat-free mass 63.8 (60.0-68.3) kg) and 12 female (triathletes, n=6; ultra-marathoners, n=6; age 38.0 ((31.6-44.5) y; height 162.0 (160.0-166.5) cm; fat mass percentage 19.4 (9.2-12.9) %; fat-free mass 45.5 (42.8-47.9) kg) ultra-endurance athletes…” (Current line: 106-109).

Line 105: Why the authors selected participants between 24-49 years? Please provide the rationale explanation. Did the authors found statistical differences between participants categories?

Answer: We agree with the expert reviewer. We targeted to apply the project on adult ultra-endurance athletes, and defined the age criteria as 20-64 years. The lower age of participation was defined by age groups of triathlon and ultra-marathon races. We defined in the criteria section the actual age range of the subjects erroneously. We corrected as follows: “Inclusion criteria for the study were: age (20-64 years old)…” (current line: 111) and “…The lower age of participation was defined by age groups of triathlon and ultra-marathon races…” (current line: 114-115). Because the age range (predominantly between 35-39 y) and training experience are quite close within the subjects, we did not find any statistical differences according to age groups compared to neither FRAP scores nor plasma parameters. Therefore, we did not discuss the paper according to age classification. However, we could add a paragraph in the conclusion part about that if you suggest to do so.

Line 108: Menstrual cycle phase was the same in all the women participants? How the authors controlled it during the experiments? Please explain carefully.

Answer: All women participants were visited the same menstrual cycle (mid-follicular) phase to eliminate the impact of menstruation on the measurements. We stated in the paper as follows: “…In the literature, the influence of menstrual cycle on endurance performance is still conflicted. Whilst some studies found no interaction between different menstrual phases and maximal anaerobic performance [39], maximal oxygen consumption, cardiorespiratory variables and blood lactate concentration [40], few studies suggested that some interactions, i.e. VO2max 2% lower in the luteal phase compared with the early follicular phase, endurance performance and muscle glycogen content were enhanced in the luteal phase. There is no consensus that suggests a specific menstrual phase for use in research on athletic performance. Therefore, to standardize the menstrual phase within subjects [41] and the estrogen effects on body antioxidant capacity by binding estrogen receptors and expand the antioxidant enzyme expressions using intracellular signal pathways [42], and to eliminate the effects of different phases on exercise performance [43], all women subjects were performed all exercise tests during mid-follicular phase (low estrogen levels, 7-9 days of the menstrual cycle). Menstrual cycle was calculated starting the first day of their period. ….” (Current line: 117-129)

Line 116: Could the authors provide more information about the time-of-day when the experiments were realized?.

Answer: We agree with the expert reviewer and added the time-of-day information at study design section as follows: “….Before the visitations, the subjects were informed to attend the laboratory at 8.00 am to 9.00 am after an overnight fast, and to refrain from exhaustive or long-duration exercise on the day preceding the visitations….” (Current line: 136)

Line 119: The authors stated "At the first visit, anthropometric parameters (body height, body weight, fat mass percentage, fat-free mass) were recorded". Please provide more information about the measurements...

Answer: We agreed with the expert reviewer and added more information about anthropometric characteristics of the study participants section. We defined in the paper as follows:”….In total, 12 male (triathletes, n=6; ultra-marathoners, n=6; age 38.5 (31.3-40.0) y; body mass 72.6 (69.6-81.0) kg; height 179.0 (173.5-184.5) cm; fat mass percentage 12.9 (10.0-16.3) %, fat-free mass 63.8 (60.0-68.3) kg) and 12 female (triathletes, n=6; ultra-marathoners, n=6; age 38.0 ((31.6-44.5) y; height 162.0 (160.0-166.5) cm; fat mass percentage 19.4 (9.2-12.9) %; fat-free mass 45.5 (42.8-47.9) kg).. ”(Current line: 106-109).

Line 162: Why the authors started the VO2max test at 5 km.h-1 and 0 % gradient on the treadmill?

Answer: We agree with the expert reviewer and realized that we did not define the test information clearly. We performed the VO2max test at 10 km.h-1 and 0 % gradient on the treadmill, however before starting the measurement, athletes warmed up at 5 km.h-1 and 0% gradient on the treadmill and after 5 min passive recovery, the VO2max test was performed. We defined in the paper as follows: “….Before the VO2max test started, five min of warm-up was performed at 5 km.h-1 and 0 % gradient on the treadmill. After 5 min of passive recovery, the incremental-running protocol started at 10 km.h-1 and with 0 % gradient with the speed increasing by 1 km.h-1 each min until a running speed of 18 km.h-1…..“ in line 181-184.

Line 164: Typo "1 km.h-1" Please correct along the manuscript.

Answer: We agree with the expert reviewer and corrected " km.h-1" as “"1 km.h-1" along the manuscript.

Line 170: The authors claim that  "BORG 6-20 Category Scale was used to measure perceived exertion after exercise". More data is needed about the time when the Borg Scale was applied.

Answer: We agree with the expert reviewer and added more information about Borg scale in the paper as follows: “….The BORG 6-20 Category Scale was used to measure RPE by asking the subjects how tough they felt during an exercise bout at the end of each stage until volitional exhaustion….” (Current line: 189-191).

Line 214: Why the authors selected 3-min and 6-min post-exercise for measuring blood lactate?.

Answer: We agree with the expert reviewer, we did not mention in paper why we prefer 3rd and 6th minutes after exercise. We added the additional information as follows: “…As our knowledge, there is no certain suggestion to determine lactate clearance after exhaustive exercise, and measurement of at the end of each 2 or 3 minutes after exercise generally preferred in literature. In addition, Gass et al. [54] conducted a study to determine blood lactate concentration following a maximal exercise in trained athletes and stated that peak lactate values after maximal exertion was reached at 6th minutes after exercise. Therefore, we preferred to measure the blood lactate concentration at the end of each 3 minutes after exercise as 3rd and 6th minutes to measure how diet in rich antioxidants effects on post-exercise lactate removal, especially at the peak lactate concentration time as it was practiced by Oh et al. [55] similarly in determining the removal of lactate after high-intensity exercise, and by Di Masi et al. [56] in comparing blood lactate clearance performed during cycling in water immersion and during cycling on land after a similar exercise bout.….” (Current line: 234-244).

Results:

Line 239: Please provide the exact p-value not p>0.05 in all the results section and provide two decimals in the effect sizes values.

Answer: We agree with the expert reviewer and provide the exact p-values in all the results section and provide two decimals in the effect sizes values.

Line 253: 16.6 (14.8-22.8) IC 95%? Please define.

Answer: The value 16.6 (14.8-22.8) is FRAP content of women subjects, and its units defined as mmol/day. (Current line: 268-269).

Line 266: The authors should change Figure 1, the figure no it so clear...

Answer: We agree with the expert reviewer and changed the Figure 1 with a high-resolution format (300 DPI). The high-resolution format also provided as material section.

Line 268: Some typos en Table 2.

Answer: We agree with the expert reviewer and corrected all the typos in Table 2.

Discussion:

Lines 304-307: Please rewording.

Answer: We agree with the expert reviewer and reworded the sentence (Current line: 355-358).

Round 2

Reviewer 2 Report

Comments to the Author

Dear Authors

Thank you for the opportunity to review this manuscript following the revision. I congratulate the authors for the detail of the updated version. Personally, I think it is much stronger paper now and is a very interesting addition to the literature in this space

Author Response

The manuscript was checked again for English spelling

all changes are marked in red